# 3D Question Answering for City Scene Understanding

## ABSTRACT

3D multimodal question answering (MQA) plays a crucial role in scene understanding by enabling intelligent agents to comprehend their surroundings in 3D environments. While existing research has primarily focused on indoor household tasks and outdoor roadside autonomous driving tasks, there has been limited exploration of city-level scene understanding tasks. Furthermore, existing research faces challenges in understanding city scenes, due to the absence of spatial semantic information and human-environment interaction information at the city level. To address these challenges, we investigate 3D MQA from both dataset and method perspectives. From the dataset perspective, we introduce a novel 3D MQA dataset named **City-3DQA** for city-level scene understanding, which is the first dataset to incorporate scene semantic and human-environment interactive tasks within the city. From the method perspective, we propose a **S**cene **g**raph enhanced **City**-level **U**nderstanding method (**Sg-CityU**), which utilizes the scene graph to introduce the spatial semantic. A new benchmark is reported and our proposed Sg-CityU achieves accuracy of 63.94% and 63.76% in different settings of City-3DQA. Compared to indoor 3D MQA methods and zero-shot using advanced large language models (LLMs), Sg-CityU demonstrates state-of-the-art (SOTA) performance in robustness and generalization. Our dataset and code are available on our project website[1].

## CCS CONCEPTS

• **Computing methodologies** → **Natural language processing**; **Scene understanding**.

## KEYWORDS

multimodal question answering, scene understanding, 3D

## 1 INTRODUCTION

City scene understanding is a crucial technology for guided tour [40], autonomous systems [15], and smart city [7]. 3D multimodal question answering (MQA) is one of the key manners of human-environment interaction to promote city scene understanding [23]. For instance, people with visual impairment could interact with the electronic personal assistant (seen as an agent) integrated into wearable smart glasses, such as Microsoft HoloLens [2] or Apple Vision Pro [1], to obtain auxiliary scenario information in the situated city by asking

---

[1]https://sites.google.com/view/city3dqa/

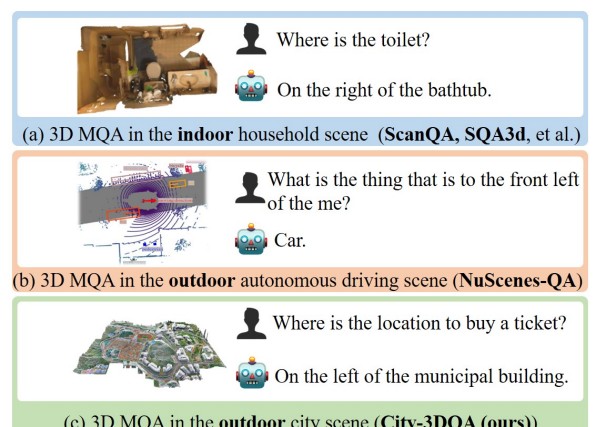

(a) 3D MQA in the **indoor** household scene (**ScanQA, SQA3d**, et al.)

Where is the toilet?

On the right of the bathtub.

(b) 3D MQA in the **outdoor** autonomous driving scene (**NuScenes-QA**)

What is the thing that is to the front left of the me?

Car.

(c) 3D MQA in the **outdoor** city scene (**City-3DQA (ours)**)

Where is the location to buy a ticket?

On the left of the municipal building.

**Figure 1: Comparison of the City-3DQA with other 3D multimodal question answering (MQA) tasks. The existing research in 3D MQA focuses on the indoor household scene (a) and outdoor autonomous driving scene (b). However, these researches lack spatial semantic and city-level interaction information within the city. City-3DQA (c) is the first dataset to focus on 3D MQA for outdoor city scene understanding.**

questions with city perception from the embedded visual sensors, shown in Figure 1 (c).

However, existing 3D MQA tasks face challenges in city scene understanding due to lacking spatial semantic information and city-level interaction information within the city, such as the location and the usage of instances. Existing research mainly focuses on two lines including the 3D MQAs in the indoor household setting (Fig. 1 (a)) and the 3D MQAs in the outdoor autonomous driving settings(Fig. 1 (b)). For the former, EQA [10], MP3D-EQA [42], MT-EQA [48] and EMQA [11] realize MQA-based scene understanding using images in indoor household scenarios through House3D simulation environment [43] for navigation tasks. Apart from using images, there is also 3D MQA research, such as 3DQA [47], ScanQA [4], CLEvR3D [45], FE-3DGQA [50] and SQA3D [28], which adopt point cloud for indoor household scene understanding based on the point cloud environment ScanNet [9]. For the latter, Qian et al. [33] introduce NuScenes-QA in outdoor settings firstly for autonomous driving using the point cloud. This task focuses on roadside-related instances including cars and pedestrians, yet it does not consider other instances in the city such as *plantings*, *buildings*, and *rivers*. In summary, current 3D MQAs are hard to satisfy city-level scene understanding for urban activities of humans or agents.

To address these challenges, we explore the task from both the dataset and method perspectives. From the dataset perspective, we introduce **City-3DQA**, the first 3D MQA dataset for outdoor city scene understanding in Figure 2. We realize data collection including City-level Instance Segmentation, Scene Semantic Extraction, and Question-Answer Pair Construction. Specifically, in City-level

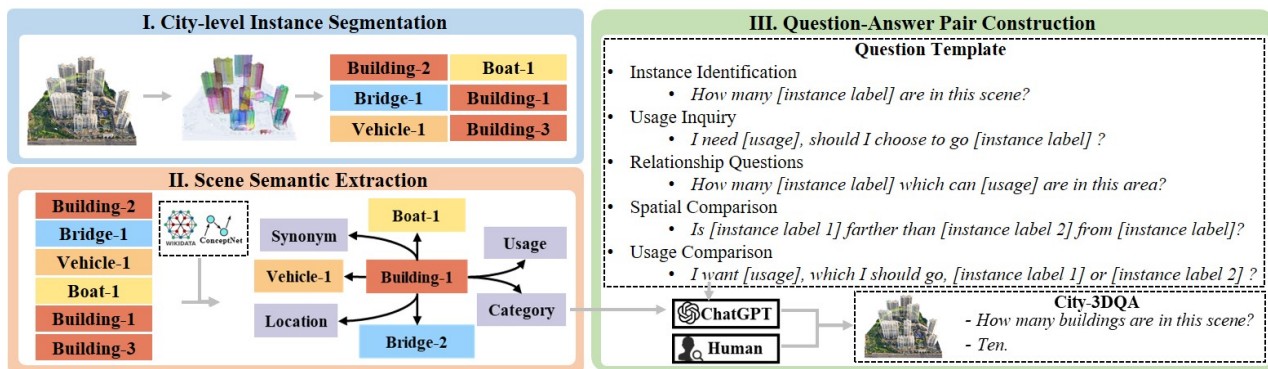

**Figure 2: Data Construction Pipeline for City-3DQA. The pipeline consists of three main stages: City-level Instance Segmentation, Scene Semantic Extraction, and Question-Answer Pair Construction.**

Instance Segmentation, we utilize pre-trained instance segmentation models to identify city instances. In Scene Semantic Extraction, we construct the scene semantic information for instances in the graph structure, including spatial information and semantic information. The spatial information denotes relationships between pairs of instances, such as "*living building - left - business building*". The semantic information represents instances with attributes, such as "*transportation building - usage - buying tickets*". In Question-Answer Pair Construction, we develop 33 unique question templates that enable multi-hop reasoning and urban activities, which are classified into five categories: instance identification, usage inquiry, relationship questions, spatial comparison, and usage comparison for the city scene understanding inspired by Gao et al. [13] and Qian et al. [33]. The LLM leverages these templates in combination with scene semantic information to produce question-answer pairs. The human evaluation assesses dataset quality. The City-3DQA dataset comprises **450k question-answer pairs** and 2.5 **billion point clouds** across six cities.

From the method perspective, we introduce a **S**cene **g**raph enhanced **City**-level **U**nderstanding method (**Sg-CityU**) for City-3DQA. Compared to indoor scene understanding, city-level scene understanding is limited by sparse semantic information due to large scales. This leads to challenges associated with long-range connections and spatial inference during the modeling process [25]. Therefore, Sg-CityU utilizes the scene graph to introduce spatial relationship information among instances. Specifically, for the input point cloud and the question, Sg-CityU extracts the vision and language representation from point clouds and questions respectively. And then a city-level scene graph is constructed, which is encoded through graph neural networks [20, 21]. We design the Fusion Layer to fuse aforementioned scene representations for answering generation.

Our main contributions can be summarized as follows:

(1) We investigate 3D multimodal question answering (MQA) to realize city-level scene understanding for urban activities of humans or agents.
(2) We introduce a novel large-scale dataset named City-3DQA. To our knowledge, City-3DQA is the first dataset to consider scene semantic information and city-level interactive tasks.

(3) We provide a baseline method (**Sg-CityU**), which introduces spatial relationship information through the scene graph to generate high-quality city-related answers.
(4) A new benchmark is proposed in which evaluations are conducted with existing MQA methods and LLM-based zero-shot methods on our City-3DQA. Experimental results show that our proposed **Sg-CityU** achieves the best performance in robustness and generalization, specifically, 63.94% and 63.76% accuracy in sentence-wise and city-wise settings respectively.

## 2 RELATED WORK

### 2.1 City Scene Understanding

Existing research in city scene understanding primarily concentrates on segmentation, reconstruction, and grounding. City segmentation, as explored in works such as Geng et al. [14], Hu et al. [17], Liao et al. [25], Yang et al. [46], aims to distinguish different instances within city-level point clouds or meshes for a comprehensive understanding of urban environments. City scene reconstruction, as discussed in Kuang et al. [22], Lin et al. [26], Tang et al. [37], Zhang et al. [49], seeks to understand the visual information of each object in city scenes and reconstruct their geometries from partial observations, such as point clouds from 3D scans. However, these methods primarily focus on visual representation rather than language representation and semantic information in city scenes, which are important for human-environment interaction. Miyanishi et al. [29] introduce CityRefer, which addresses city-level visual grounding by localizing objects in 3D scenes based on language expressions. Inspired by these studies, our research aims to tackle this problem from a multimodal question answering perspective. We propose the first 3D multimodal question answering dataset, City-3DQA, for 3D city scene understanding, which integrates language representation and semantic information.

### 2.2 3D Multimodal Question Answering

3D Multimodal Question Answering is a novel task within the field of scene understanding, concentrating on the ability to answer questions about 3D scenes, which are depicted through simulated environments or point clouds [4]. Das et al. [10], Datta et al.

[11], Wijmans et al. [42], Yu et al. [48] present an embodied question answering where the agent must first intelligently navigate to explore the environment, gather the necessary visual information through first-person vision, and then respond to the question in a 3D simulated environment. Azuma et al. [4], Etesam et al. [12], Ma et al. [28], Yan et al. [45], Ye et al. [47], Zhao et al. [50] propose a series of studies based on the ScanNet dataset [9] that focus on processing point cloud data from entire 3D indoor scenes to respond to specific textual queries about the environment. However, these works focus on the indoor household scene and overlook the outdoor scene. Qian et al. [33] proposes the outdoor 3D multimodal question NuScenes-QA answering benchmark to address the human-machine interaction in autonomous driving rather than the city scene understanding. We first introduce City-3DQA, a 3D question-answering dataset specifically designed for the understanding of outdoor city scenes. Unlike the NuScenes-QA which concentrates on roadside areas, City-3DQA emphasizes the comprehension of city landscapes along with their spatial characteristics. Additionally, it incorporates features related to interaction, such as usage.

## 3 PROBLEM DEFINITION

The 3D MQA for city scene understanding is formulated as follows: given inputs of the point cloud $p$ and question $q$ about the 3D city scene, the model aims to output $\widehat{a}$ that semantically matches true answer $a^*$ from the answer set $\mathbb{A}$,

$$\widehat{a} = \arg\max_{a \in \mathbb{A}} \ P(a|p, q). \tag{1}$$

Understanding city-level scenes is more challenging than indoor scenes. This is because city scenes have less dense information over large areas, making it hard to model long-range connections and spatial relationships [25]. Therefore, we introduce a scene graph $sg$ which contains the relative spatial relationship [44]. The $sg$ is composed of nodes and edges, where the nodes represent instances and the edges represent the spatial relationships between these instances. We consider a scene-graph-aware joint probability model for the task using $sg$ and decompose Equation 1 into two parts, given by:

$$P(a|p, q) = P(a|p, q, sg) \times P(sg|p). \tag{2}$$

## 4 CITY-3DQA DATASET

### 4.1 Data Construction

We develop an automatic pipeline for the construction of the City-3DQA dataset, as depicted in Figure 2. The City-3DQA dataset is derived from the 3D city point cloud dataset UrbanBIS [46]. Our pipeline encompasses three primary components: City-level Instance Segmentation, Scene Semantic Extraction, and Question-Answer Pair Construction.

**City-level Instance Segmentation.** We use pre-trained instance segmentation [46] for the UrbanBIS dataset and obtain a wide range of city instances including buildings, vehicles, vegetation, roads, and bridges covering six cities, Qingdao, Wuhu, Longhua, Yuehai, Lihu, and Yingrenshi. We extract the instance-level label along with annotations and spatial locations to build the instance set $S_I = \{i, (x_i, y_i, z_i)|i \in I\}$ from UrbanBIS, where $I$ is the instances

from the raw dataset. $x_i, y_i, z_i$ is the x-axis, y-axis, and z-axis coordinate for each $i$.

**Scene Semantic Extraction.** We construct the scene semantic information $G_i$ for each instance $i$ in the graph structure, which comprises two components: the spatial information $sp_i$ and the semantic information $se_i$ in the graph structure. $sp_i$ contains a series triples $(i, r_{i,j}^{sp}, j)$, where $r_{i,j}^{sp}$ is the spatial relationship between the instances $(i, j)$, where $i \in S_I, j \in S_I$. These relationships are centered around instance $i$ and we define counterclockwise as the positive direction. $R_{i,j}$ are divided via eight relationships: "*front*", "*front-right*", "*right*", "*back-right*", "*front-left*", "*left*", "*back-left*" and "*back*", depending on relative instance spatial positions and the angle between instance $i$ and $j$,

$$\theta = \arctan \frac{y_j - y_i}{x_j - x_i},$$

$$r_{i,j}^{sg} = \begin{cases} \text{front} & \text{if} -22.5° < \theta \leq 22.5° \\ \text{front-right} & \text{if } 22.5° < \theta \leq 67.5° \\ \text{right} & \text{if } 67.5° < \theta \leq 112.5° \\ \text{back-right} & \text{if } 112.5° < \theta \leq 157.5° \\ \text{front-left} & \text{if} -67.5° < \theta \leq -22.5° \\ \text{left} & \text{if} -112.5° < \theta \leq -67.5° \\ \text{back-left} & \text{if} -157.5° < \theta \leq -112.5° \\ \text{back} & \text{else .} \end{cases} \tag{3}$$

$se_i$ are defined as triples $(i, r_i^{se}, v_i)$, where $r_i^{se}$ and $v_i$ are the attribute and value for instance $i$ respectively. In City-3DQA, we define $r_i^{se}$ as five attributes including instance label, building category label, synonym label, location, and usage label. The instance label and a detailed building category label are sourced from the pre-trained instance segmentation method [46]. Drawing inspiration from Henderson et al. [16], we acknowledge the usage label as an important aspect of urban activities within the city scene. To enhance the relevance of the City-3DQA datasets to a common language and to promote linguistic variety, we integrate synonyms, as suggested by [35]. The sources for usage descriptions and synonym labels are knowledge base WikiData [39] and ConceptNet [36].

**Question-Answer Pair Construction.** To construct the question-answer pairs automatically, we propose a template-based pipeline utilizing LLM to transform structured data $G_i$ into unstructured language question $q_i$ and answer $a_i$ for the instance $i$. In our study, we formulate two distinct questions using the $G_i$ within the City-3DQA framework. The first question aims to extract the tail $j$ in $sp_i = \{i, r_{i,j}^{sp}, j\}$ or the value $v_i$ in $se_i = \{i, r_i^{se}, v_i\}$, to build the answer in the question-answer pair. The second question concentrates on identifying the edge between the tail and head of a triplet, such as the relationship $r_{i,j}^{sp}$ in $sp_i = \{i, r_{i,j}^{sp}, j\}$ or the attribute $r_i^{se}$ in $se_i = \{i, r_i^{se}, v_i\}$, to formulate the answer in the question-answer pair.

Building upon the work of Gao et al. [13] and Qian et al. [33], the City-3DQA dataset is comprised of 33 question templates, which are categorized into five categories: instance identification, usage inquiry, relationship questions, spatial comparison, and usage comparison. These templates are detailed in the supplementary material. The first three categories of templates are designed to evaluate the presence, quantity, and characteristics of instances within city

**Table 1: Comparison between City-3DQA and other 3D MQA datasets. Question-Answer Pairs and Point Clouds denote the number of question-answer pairs and points.**

| Dataset | Scene | Collection | Scale | Input Modal | Question-Answer Pairs | Point Clouds |
|---|---|---|---|---|---|---|
| EQA [10] | indoor | template | Room | image | 1.5k | - |
| MP3D-EQA [42] | indoor | template | Room | image | 1.1k | - |
| EMQA [11] | indoor | human | Room | image | 9.7k | - |
| MT-EQA [48] | indoor | template+human | Room | image | 19k | - |
| 3DVQA [12] | indoor | template | Room | point cloud | 484k | 242M |
| 3DQA [47] | indoor | human | Room | point cloud | 10k | 242M |
| ScanQA [4] | indoor | auto + human | Room | point cloud | 41k | 242M |
| CLEVR3D [45] | indoor | template | Room | point cloud | 60.1k | 242M |
| FE-3DGQA [50] | indoor | human | Room | point cloud | 20k | 242M |
| SQA3D [28] | indoor | human | Room | point cloud + image | 33.4k | 242M |
| NuScenes-QA [33] | outdoor | template | Roadside | point cloud + image | 460k | 1.4B |
| **City-3DQA (ours)** | outdoor | template + auto + human | City | point cloud | 450k | 2.5B |

scenes, including their usages and relationships and urban activities. These templates necessitate straightforward answers and are categorized as single-hop questions. For example, questions such as "*What is the usage of [instance label]?*" and "*Where is [instance label]?*" are formulated. To facilitate the construction of these questions, we employ slots like "*[instance label]*", "*[location]*", and "*[usage]*" for completion by LLMs. The last two categories of templates are designed to evaluate the comparison of instances within city scenes, including their usages and relationships. These templates necessitate a multi-hop reasoning step to arrive at the answer and they are classified into multi-hop questions For instance, inquiries such as "*I want [usage], which I should go, [instance label 1] or [instance label 2] ?*" and "*Between [instance label 1] and [instance label 2], which is nearest to [instance label]?*" are devised. We utilize slots such as "*[instance label 1]*" and "*[instance label 2]*" in the templates for the comparative analysis of instances in the city.

We design the prompt function $f_{prompt}(\cdot)$ which incorporates slots. The details of the prompt are shown in the supplementary material. These slots are populated using the input $G_i$. We utilize the ChatGPT API with the gpt-3.5-turbo model. The whole pipeline can be formulated as below:

$$(q_i, a_i) = search\ \text{LLM}(f_{prompt}(G_i)), \qquad (4)$$

where the search function $search(\cdot)$ could be an argmax function that searches for the highest-scoring output or sampling that randomly generates outputs following the probability distribution of the adopted LLM. The prompt engineering $f_{prompt}(\cdot)$ is detailed in the supplementary material. The LLM combination with templates offers linguistic diversity and improves the quality of the corpus compared to using templates alone [41]. After the automated generation of question-answer pairs by LLMs, we conduct the human evaluation to assess and guarantee the quality and accuracy of the City-3DQA dataset.

### 4.2 Data Statistics

In the vision modal, City-3DQA covers 193 unique city scenes across six cities including Qingdao, Wuhu, Longhua, Yuehai, Lihu, and Yingrenshi, incorporating **2.5 billion** point clouds. The combined coverage of these scenes extends over an area of 10.78 square kilometers. The dataset includes information from $3,370$ instances of various city instances such as buildings, bridges, vehicles, and boats. The comparison between City-3DQA and other 3D MQA works is shown in Table 1.

In the language modal, the City-3DQA dataset comprises **450k** question-answer pairs covering five different questions in city scene understanding including instance identification, usage inquiry, relationship questions, spatial comparison, and usage comparison. Figure 3 illustrates the basic statistics of our dataset of language modal. In Figure 3(a), the distribution of question types in the dataset is as follows: usage inquiry (5.6%), instance identification (6.3%), relationship question (35.3%), spatial comparison (32.5%), and usage comparison (20.3%). Furthermore, the dataset comprises 47.2% single-hop questions and 52.8% multi-hop questions. Figure 3(b) demonstrates that the lengths of our questions vary significantly, ranging from five to twenty-five words. Figure 3(c) presents a visualization of the extensive vocabulary employed in the questions of our dataset.

## 5 METHOD

We propose a framework to model Equation 2, named **Sg-CityU** (**S**cene **g**raph enhanced **City**-level **U**nderstanding) method shown in Figure 4 (a). Sg-CityU model consists of Multimodal Encoder, Fusion Layer, and Answer Layer.

### 5.1 Multimodal Encoder

We use the input point cloud $p$ consisting of point coordinates $c \in \mathbb{R}^3$ in the 3D space for 3D representation. Following previous 3D and language research, we use additional point features such as the height of the point, colors, and normals [4, 8]. Sg-CityU detects objects in the scene based on point cloud features using VoteNet [31], which uses PointNet++ [32] as a backbone network. We get object proposals from VoteNet for the instances and the whole scan and project them through the multi-layer perceptron (MLP) to obtain the object proposal representation,

$$F_p = \text{MLP}(\text{VoteNet}(i_p)), \qquad (5)$$


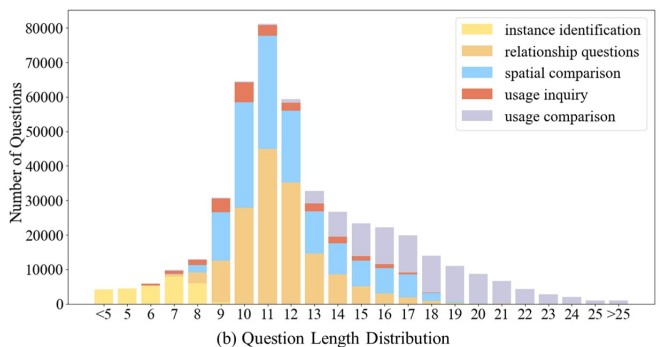
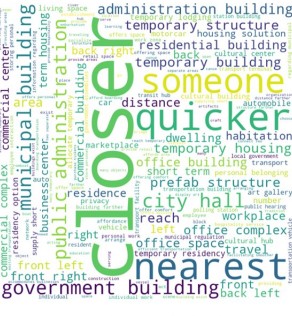

(a) Question Distribution                          (b) Question Length Distribution                          (c) Question Word Cloud

**Figure 3: The statistical distributions of questions within the City-3DQA dataset are presented. The question length means the number of words in the question sentence. Multi and Single mean the multi-hop questions and single-hop questions respectively.**

where $F_p \in \mathbb{R}^{dim \times N}$ and $i_p$ is the point cloud for the instances. $dim$ represents the hidden size of representation, and $n$ indicates the number of proposals. A question sentence $q$ is fed to the pre-trained language model encoder BERT [19] and MLP to calculate the question features $F_q \in \mathbb{R}^{dim}$,

$$F_q = \text{MLP}(\text{BERT}(q)). \tag{6}$$

We construct the $sg$ based on $i_p$ to introduce spatial relationship among $i_p$. The $sg$ comprises nodes, which represent instances, and edges, which denote the spatial relationships between these instances. The relationships are divided and defined as Equation 3. We encode $sg$ through $n$-layers graph convolutional networks (GCN) [21] and output the representation $F_{sg} \in \mathbb{R}^{dim \times N}$,

$$
\begin{aligned}
sg^{m+1} &= \text{GCN}^m(sg^m), \\
F_{sg} &= \text{MLP}(sg^{m+1}),
\end{aligned} \tag{7}
$$

where $GCN^m$ is the learnable GCNs at the $m$-th layer, and $F_{sg}$ is the feature of the node after encoding by $m$-th GCN layer. Inspired by language model type condition [24], we initialize $sg^0$ with the word embeddings of the nodes and edges.

### 5.2 Fusion Layer

In the Fusion Layer, we design the multimodal fusion network (MMFN) for the different inputs as shown in Figure 4 (b). Specifically, MMFN consists of self-attention and cross-attention and takes $F_p$, $F_q$, $F_{sg}$ as input,

$$
\begin{aligned}
F_q &= \text{Self-Attention}(F_q), \\
F_p &= \text{Self-Attention}(F_p), \\
F_p &= \text{Cross-Attention}(F_p, F_q), \\
F_{sg} &= \text{Self-Attention}(F_{sg}), \\
F_{sg} &= \text{Cross-Attention}(F_p, F_{sg}),
\end{aligned} \tag{8}
$$

We perform the fusion multimodal features through the Fusion Layers consisting of $l$-th MMFN layer cascaded in depth,

$$F_p^l, F_q^l, F_{sg}^l = \text{MMFN}^l(F_p^{l-1}, F_q^{l-1}, F_{sg}^{l-1}), \tag{9}$$

For $\text{MMFN}^0$, we set its input features $F_p^0 = F_p$, $F_q^0 = F_q$, $F_{sg}^0 = F_{sg}$, respectively.

**Table 2: Different split in City-3DQA. It denotes the number of question-answer pairs and cities in different set in the split mode.**

| Split | train | | | val | | | test | | |
|---|---|---|---|---|---|---|---|---|---|
| | Single | Multi | All | Single | Multi | All | Single | Multi | All |
| Sentence-wise | 173k | 136k | 310k | 34k | 44k | 78k | 35k | 26k | 61k |
| City-wise | 176k | 133k | 310k | 37k | 41k | 78k | 35k | 26k | 61k |

### 5.3 Answer Layer

We map the fused features to the answer set $\mathbb{A}$ that matches the true answer for answer prediction with MLP,

$$F_f = \text{MLP}(\text{Concat}(F_p^l, F_q^l, F_{sg}^l)), \tag{10}$$

where $\text{Concat}(\cdot)$ is the concatenation and $F_f \in \mathbb{R}^{dim_A \times dim}$, $dim_A$ is the dimension of the answer set $\mathbb{A}$. To consider multiple answers, we compute final scores with the cross-entropy (CE) loss function to train the module.

## 6 EXPERIMENT

### 6.1 Implementation Details

**Data Organization.** To train and evaluate our proposed models, we split our City-3DQA dataset using two different modes: sentence-wise and city-wise. In the city-wise split, we categorize the examples by city. This results in four cities (Longhua, Wuhu, Qingdao, Yingrenshi) being allocated to the training set, one city (Lihu) to the validation set, and one city (Yuehai) to the test set. For the sentence-wise split, we divide the 450K question-answer pairs in City-3DQA into training, validation, and test sets with the same ratio as the city-wise split respectively and each set contains the six cities. The distribution of examples in each set, according to these splits, is detailed in Table 2.

**Training Details.** We employ the Adam optimizer with weight decay $5e^{-4}$, a learning rate of $1e^{-3}$, and a batch size of 50 during the training stage. Experiments are implemented with CUDA 11.2 and PyTorch 1.7.1 and run on an NVIDIA RTX A6000.

**Metrics.** We adopt the Top-1 accuracy (Top@1) and Top-10 accuracy (Top@10) as our evaluation metric, following the practice of

Figure 4: The framework of our proposed model Sg-CityU (a) and Fusion Layer in Sg-CityU (b). In Sg-CityU, the question, scene graph, and point clouds are processed by the feature extraction backbone to obtain multimodal features. Finally, the multimodal features are fed into Fusion Layer and Answer Layer for answer generation. In Fusion Layer, we build layers of multimodal fusion network (MMFN) based on self-attention and cross-attention to fuse different model inputs.

many other MQA methods [3, 4], and evaluate the performance of different question types separately.

**Baselines.** We design two categories of baselines for comparison in City-3DQA:

- **General LLMs.** We utilize LLM as baselines into two types: multimodal LLM utilizing 2D images and LLM utilizing scene graphs as input. For the former, we convert the input point clouds into 2D images. This process ensures alignment with the requirements of multimodal LLMs using 2D image input following Ma et al. [28]. Our selected baselines for this category include Qwen-VL [6], and LLaVA [27]. For the latter, we construct the scene graph from each city scene and we organize these scene graphs in language. Our selected baselines for this category include Qwen [5], and Llama-2 [38]. LLMs generate answers based on the questions and input and we select the most similar answers from answer spaces $\mathbb{A}$ based on the BERT score [34]. The prompt engineering used in LLM evaluation is detailed in supplementary material.

- **Indoor Models.** We choose the baseline models ScanQA, CLIP-Guided, 3D-VLP, and the state-of-the-art (SOTA) model 3D-VisTA using in indoor 3D MQA datasets ScanQA [4] and transfer it from indoor setting into outdoor setting. These models take point cloud as input and our model Sg-CityU takes point cloud and scene graph as input.

## 6.2 Results Analysis

*6.2.1 **Comparison with General LLMs.*** We compare our proposed models with the LLMs in zero-shot setting in Table 3 and our proposed model Sg-CityU outperforms in all metrics. For multimodal LLM using the projection image as input, Qwen-VL [6] demonstrates the acc@1 of 18.81% and 19.75% across all sets for sentence-wise and city-wise evaluation, respectively. Furthermore, it achieves the acc@10 of 63.86% and 63.71% in the same respective categories. On the other hand, LLaVA [27] attains an acc@1

of 20.60% and 20.56% for sentence-wise and city-wise evaluation, respectively, and an acc@10 of 67.37% and 67.02% in the corresponding test sets. Compared to the best results in multimodal LLM, Sg-CityU achieves more than 3.1 times improvement in sentence-wise (20.60% → 63.94%) and city-wise (20.56% → 63.76%) in acc@1 and 1.4 times improvements in sentence-wise (67.37% → 98.81%) and city-wise (67.02% → 98.68%) in acc@10. We attribute the poor performance of multimodal LLM to two points. First, in the zero-shot setting of multimodal LLMs, there is a lack of parameters to bridge the domain gap between the pre-trained domain and the City-3DQA domain through fine-tuning. Second, the projection image fails to accurately represent the city scene in point cloud.

For LLM using the scene graph as input, Qwen [5] achieves 30.35% and 31.31% of acc@1 in sentence-wise and city-wise, 73.84% and 75.26% of acc@10 in sentence-wise and city-wise. Llama-2 [38] achieves 37.66% and 38.37% of acc@1 in sentence-wise and city-wise, 80.02% and 79.34% of acc@10 in sentence-wise and city-wise. Compared to multimodal LLMs, LLMs with scene graphs achieve better performance and we attribute it to the LLM generalization performance in the language. Compared to the best results in LLM, Sg-CityU achieves more than 20% points improvement in sentence-wise (37.66% → 63.94%) and city-wise (38.37% → 63.76%) in acc@1 and over 10% points improvements in sentence-wise (80.02% → 98.81%) and city-wise (79.34% → 98.68%) in acc@10. The suboptimal performance of LLMs can be attributed to two points. First, due to the context window length restriction, the language input based on the scene graph can only cover part representation, constraining the understanding of the city-level scene. In a city scene comprising $n$ instances, the corresponding scene graph contains $\frac{n(n+1)}{2}$ triples. The context windows of Llama-2 and Qwen are $4k$ and over 25% input sentences with scene graphs are over the the window sizes. Second, LLMs overlook the visual features present in city scenes, which are beneficial for the performance of 3D MQA tasks.

Table 3: The comparison between our model and different methods. We compare eight different methods with Sg-CityU and Sg-CityU achieves the best score in all metrics compared to the methods. The scene graphs are organized as language.

| Category | Models | Input | Sentence-wise | | | | | | City-wise | | | | | |
|---|---|---|---|---|---|---|---|---|---|---|---|---|---|---|
| | | | Single-hop | | Multi-hop | | All | | Single-hop | | Multi-hop | | All | |
| | | | acc@1 | acc@10 | acc@1 | acc@10 | acc@1 | acc@10 | acc@1 | acc@10 | acc@1 | acc@10 | acc@1 | acc@10 |
| General LLMs | Qwen-VL [6] | Image | 30.53 | 70.85 | 9.76 | 58.45 | 18.81 | 63.86 | 30.79 | 71.07 | 9.78 | 57.07 | 19.75 | 63.71 |
| | LLaVA [27] | Image | 33.93 | 77.02 | 10.33 | 59.92 | 20.60 | 67.37 | 32.56 | 76.94 | 9.84 | 58.07 | 20.56 | 67.02 |
| | Qwen [5] | Scene Graph | 55.25 | 85.41 | 11.21 | 63.48 | 30.35 | 73.84 | 55.40 | 85.49 | 12.59 | 66.35 | 31.31 | 75.26 |
| | Llama-2 [38] | Scene Graph | 60.51 | 86.34 | 20.00 | 75.13 | 37.66 | 80.02 | 60.03 | 86.18 | 18.82 | 73.17 | 38.37 | 79.34 |
| Indoor Models | ScanQA [4] | Point Cloud | 76.42 | 90.75 | 28.31 | 86.46 | 49.28 | 88.34 | 64.84 | 88.73 | 27.03 | 84.37 | 47.33 | 86.45 |
| | CLIP-Guided [30] | Point Cloud | 74.54 | 98.49 | 33.73 | 97.54 | 51.55 | 98.38 | 63.05 | 98.35 | 32.41 | 97.12 | 46.94 | 98.00 |
| | 3D-VLP [18] | Point Cloud | 72.78 | 98.55 | 35.54 | 97.76 | 51.72 | 98.40 | 64.03 | 98.42 | 34.95 | 97.19 | 48.74 | 98.33 |
| | 3D-VisTA [51] | Point Cloud | 79.23 | 98.52 | 44.67 | 97.85 | 59.63 | 98.37 | 71.28 | 98.47 | 43.87 | 97.56 | 56.74 | 98.48 |
| | **Sg-CityU (ours)** | Point Cloud + Scene Graph | **80.95** | **98.86** | **50.75** | **98.66** | **63.94** | **98.81** | **78.46** | **98.76** | **50.50** | **98.45** | **63.76** | **98.68** |

*6.2.2* **Comparison with Indoor Models.** We conduct the comparative experiments between Sg-CityU and models in indoor settings shown in Table 3. For SOTA model 3D-VisTA [4] in the indoor setting, Sg-CityU achieves 4.31% points improvement in sentence-wise (59.63% → 63.94%) and 7.02% points improvement city-wise (56.74% → 63.76%) in acc@1 and 0.44% points improvements in sentence-wise (98.37% → 98.81%) and 0.20% points improvements city-wise (98.48% → 98.68%) in acc@10. Compared to indoor MQA models, the efficiency of Sg-CityU is attributed to the scene graph, which offers a semantic and spatial representation of city-level outdoor scenes. This representation features sparse instances that encompass a wide range of city-level scenes.

To evaluate the generalization and robustness of indoor models and Sg-CityU in diverse city scenes, our research includes a comparative analysis of their performance across different cities. In this study, we assess the performance of the models used in indoor settings and Sg-CityU models under two different settings: city-wise and sentence-wise. In the city-wise evaluation, ScanQA achieves an accuracy of 47.33% for acc@1 and 86.45% for acc@10. These figures represent a decline in performance compared to the sentence-wise setting, where acc@1 decreases by 1.95% (49.28% → 47.33%) and acc@10 decreases by 1.89% (88.34% → 86.45%). Similar trends are observed in other indoor MQA models, with CLIP-Guided experiencing a decrease of 4.61% (51.55% → 46.94%), 3D-VLP a decrease of 2.98% (51.72% → 48.74%), and 3D-VisTA a decrease of 2.89% (59.63% → 56.74%). In contrast, Sg-CityU shows a decline of 0.18% in acc@1 (63.94% → 63.76%) and 0.13% in acc@10 (98.81% → 98.68%) when comparing the city-wise to the sentence-wise setting. These results show that our model exhibits generalization and robustness capabilities across diverse city-level scenes compared to the indoor models.

*6.2.3* **Comparison in Multi-hop Questions.** We conduct experiments on both multi-hop and single-hop questions, comparing the performance of baseline models and the proposed Sg-CityU model, as presented in Table 3. Our findings show that the multimodal LLMs with image input exhibit suboptimal performance in multi-hop questions, with an acc@1 of 10.33% and 9.84% in sentence-wise and city-wise evaluations, respectively, for LLaVA, and 9.76% and 9.78% for Qwen-VL. LLMs utilizing scene graphs demonstrate superior performance, with Qwen achieving 11.21% and 12.59% in sentence-wise and city-wise evaluations, respectively, and Llama-2

achieving 20.00% and 18.82%. However, supervised models achieve better performances. In multi-hop questions, ScanQA achieves 8.31% (20.00% → 28.31%) improvements in sentence-wise and 8.21% (18.82% → 27.03%) improvements in city-wise compared to the best performance of general LLM. CLIP-Guided shows a 13.73% (20.00% → 33.73%) improvement in sentence-wise accuracy and a 13.59% (18.82% → 32.41%) improvement in city-wise accuracy. 3D-VLP achieves a 15.54% (20.00% → 35.54%) improvement in sentence-wise accuracy and a 16.13% (18.82% → 34.95%) improvement in city-wise accuracy. 3D-VisTA exhibits a 24.67% (20.00% → 44.67%) improvement in sentence-wise accuracy and a 25.05% (18.82% → 43.87%) improvement in city-wise accuracy. Similarly, our model Sg-CityU achieves an improvement of 30.75% (20.00% → 50.75%) in sentence-wise accuracy and 31.68% (18.82% → 50.50%) in city-wise accuracy compared to the best performance of general LLMs. We attribute this limitation to the domain gap between the training datasets of LLMs and the requirements for understanding city scenes. Therefore, LLMs cannot comprehend visual features in point clouds and the scene graph at the city level.

*6.2.4* **Ablation Study in Sg-CityU.** We conduct an ablation study to evaluate the effect of the scene graph on the performance of our proposed method Sg-CityU, as detailed in Table 4. When employing the scene graph as the input alone, Sg-CityU achieves 31.48% and 29.00% of acc@1 in sentence-wise and city-wise, 96.45% and 95.77% of acc@10 in sentence-wise and city-wise. These results indicate that Sg-CityU relying on the scene graph alone as input can not yield optimal performance and we attribute the absence of visual features. When utilizing the point cloud as the input alone, Sg-CityU achieves 52.25% and 49.01% of acc@1 in sentence-wise and city-wise, 98.07% and 97.40% of acc@10 in sentence-wise and city-wise. When employing the scene graph as assistance, Sg-CityU achieves 11.69% points improvement in sentence-wise (52.25% → 63.94%) and 14.75% points improvement city-wise (49.01% → 63.76%) in acc@1 and 0.61% points improvements in sentence-wise (98.07% → 98.68%) and 1.41% points improvements city-wise (97.40% → 98.81%) in acc@10. Incorporating scene graphs into the framework can enhance the performance of our proposed method Sg-CityU in City-3DQA tasks. This improvement is achieved by providing a more structured representation of city-level scenes, which facilitates an understanding of the spatial and semantic relationships between various instances.

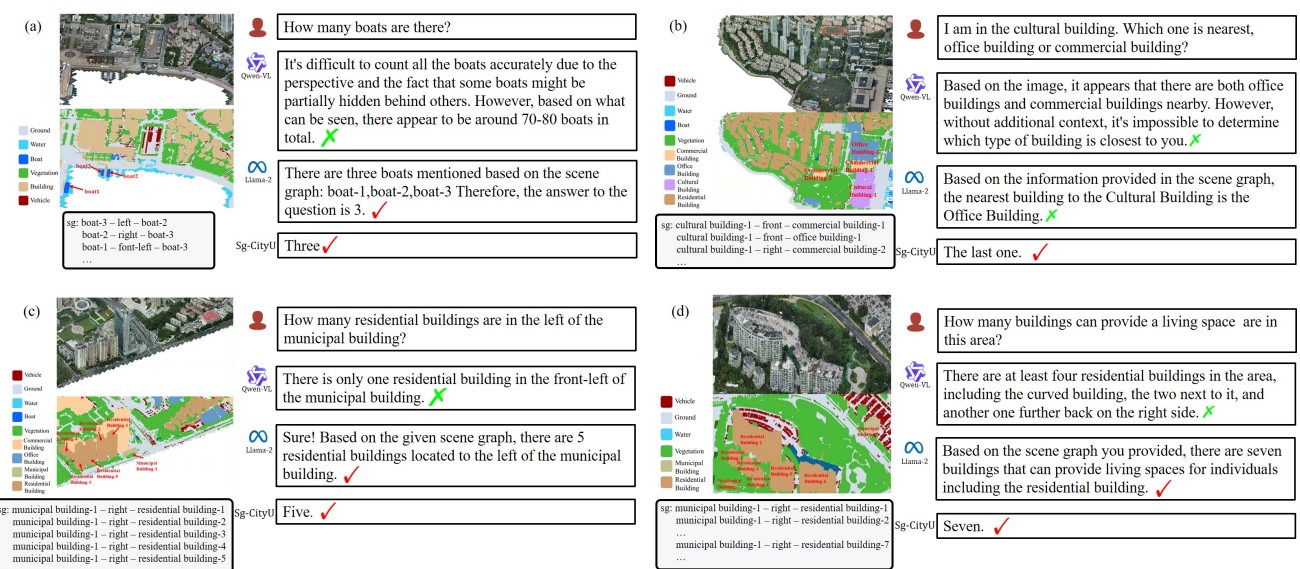

**Figure 5: Visualization of examples. We compare and visualize the answer generated by Qwen-VL, Llama-2 and Sg-CityU. We visualize the city scene with the instance label and scene graph (sg). ✓ and ✗ mean the correct answer and wrong answer respectively.**

**Table 4: Ablation study on the input modal of Sg-CityU. This study specifically examines the effects of removing the point cloud and scene graph inputs while retaining the question input.**

| Input | | | Sentence-wise | | City-wise | |
|---|---|---|---|---|---|---|
| Question | Scene Graph | Point Cloud | acc@1 | acc@10 | acc@1 | acc@10 |
| ✓ | ✗ | ✓ | 52.25 | 98.07 | 49.01 | 97.40 |
| ✓ | ✓ | ✗ | 31.48 | 96.45 | 29.00 | 95.77 |
| ✓ | ✓ | ✓ | **63.94** | **98.68** | **63.76** | **98.81** |

## 6.3 Visualization and Case Study

We randomly select the cases and visualize them in Figure 5. In each case, we present the following components: the posed question, the scene with instance labels, and the corresponding scene graph. We compare the answers generated by three different models: the language-only LLM (Llama-2), the multimodal LLM (Qwen-VL), and the Sg-CityU model trained sentence-wise.

In **Case (a)**, we present the question, "*How many boats are there?*" Qwen-VL produces inaccurate answers due to a domain gap between its training datasets, which consist of 2D images sourced from the Internet, and the 3D point cloud images it encounters in the application. This gap leads to hallucinated answers. In contrast, Llama-2 based on the scene graph and Sg-CityU comprehends this city scene. In **Case (b)**, we pose the question, "*I am in the cultural building. Which one is nearest, the office building or commercial building?*" Both Qwen-VL and Llama-2 generate incorrect answers. We attribute this to the deficiency in the LLM's understanding of geographic scale information within the visual features. Scene graphs used in LLMs lack information regarding the distances between instances, leading to hallucinated answers. In **Case (c)**, we investigate the query, "*How many residential buildings are located to the left of the municipal building?*". Llama-2 generates accurate responses, whereas Qwen-VL generates incorrect ones. We attribute it to the fact that LLMs based on scene graphs can leverage the relative spatial position within a scene graph for specific instances. In contrast, multimodal LLMs cannot comprehend the concept of "*left*" within the city scene using projection 2D images. In **Case (d)**, we pose the question, "*How many buildings can provide a living space in this area?*" Qwen-VL can detect the curved building as a residential building however, it can not detect the other dense and small residential buildings, leading to incorrect answers.

## 7 CONCLUSION

In this work, we investigate the 3D multimodal question answering (MQA) task for city scene understanding from both dataset and method perspectives. Firstly, we introduce a large-scale dataset, **City-3DQA**, designed to encompass a wide range of urban activities, facilitating enhanced comprehension at the city level. Secondly, a scene graph enhanced city scene understanding method **Sg-CityU** is proposed to deal with the long-range connections and spatial inference challenges in city-level scene understanding. Experiments show that our proposed method outperforms the indoor MQA models and the large language models, showing robustness and generalization across different cities. To our knowledge, we are the first to explore the 3D MQA task for the city scene understanding in both the dataset and method aspects, which can promote the development of human-environment interaction within cities.

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
