# OpenReview forum: "3D Question Answering for City Scene Understanding"
_acmmm.org/ACMMM/2024/Conference — MM2024 Poster_

### Official Review · Reviewer_Free · 2024-05-21

**Rating:** 4
**Confidence:** 3

**Summary:**

The paper presents a new 3D MQA dataset for city-level scene understanding and introduces a scene-graph enhanced method to address the city-level 3D MQA task.
The proposed approach is both reasonable and applicable. Experimental results demonstrate the effectiveness of the method.

**Strengths:**

1. The dataset construction process and the proposed method are presented clearly and are easy to follow.
2. The experimental results demonstrate good performance on the proposed dataset.

**Limitations:**

1. Inappropriate Expression:
   - Line 286 states, "covering six cities, Qingdao, Wuhu, Longhua, Yuehai, ..." should be revised to "covering six cities (areas)," as the latter four are districts in Shenzhen, not standalone cities.

2. Lack of Discussion on Related Work:
   - The paper lacks a thorough discussion of related work concerning 3D scene graph construction. Works such as 3DSSG[1] and VL-SAT[2] also address this topic, and their insights should be incorporated into the related work section.

3. Confusion in Figure 4:
   - Although Line 490 indicates that the scene graph construction relies on instance predictions from VoteNet, Figure 4 depicts scene graph construction from the raw input point cloud. This inconsistency requires clarification.

4. Answer Layer Implementation:
   - Line 556 mentions an "answer set," suggesting that the answer process is formulated as a multi-class classification problem rather than an auto-regressive approach commonly used in LLM. This classification setting constrains the model's output space.

5. Generalizability Concerns:
   - The proposed method appears applicable to other datasets like ScanQA. However, it remains unclear whether the method or network architecture will perform well on different datasets.

[1] Wald J, Dhamo H, Navab N, et al. Learning 3d semantic scene graphs from 3d indoor reconstructions[C]//Proceedings of the IEEE/CVF Conference on Computer Vision and Pattern Recognition. 2020: 3961-3970.

[2] Wang Z, Cheng B, Zhao L, et al. VL-SAT: visual-linguistic semantics assisted training for 3D semantic scene graph prediction in point cloud[C]//Proceedings of the IEEE/CVF Conference on Computer Vision and Pattern Recognition. 2023: 21560-21569.

**Suitability:**

3

---

### Official Review · Reviewer_qAWN · 2024-05-22

**Rating:** 4
**Confidence:** 3

**Summary:**

This paper introduces a novel approach to 3D multimodal question answering (MQA) for urban environments, expanding beyond the commonly studied indoor household and autonomous driving scenarios. The authors identify a gap in comprehensive city-level scene understanding and propose a new dataset, City-3DQA, alongside a novel method, Sg-CityU, to address these challenges.

**Strengths:**

(1)The paper is well-structured and clearly written, making it easy to follow the authors' contributions and findings. The figures and tables are informative and support the text effectively.
(2)The paper introduces the first dataset and method specifically designed for 3D multimodal question answering at the city level.
(3)The City-3DQA dataset is extensive, encompassing 450,000 question-answer pairs and 2.5 billion point clouds over six cities.
(4)The introduction of the City-3DQA dataset and the Sg-CityU method has potential applications in various fields, such as urban planning, autonomous navigation, and assistive technologies for visually impaired individuals.

**Limitations:**

(1)The City-3DQA dataset is constructed using an automatic pipeline that involves instance segmentation, scene semantic extraction, and question-answer pair generation using language models. While the authors provide some details on the data construction process, there is limited discussion on the potential biases that may be introduced during this process, such as biases in the instance segmentation models or the language models used for question-answer generation.
(2)The experiments are conducted on the City-3DQA dataset, which covers six cities.  However, the paper does not provide a thorough discussion on the scalability of the proposed method to larger and more diverse urban environments. Additionally, the generalization ability of the Sg-CityU method to other city-level tasks, such as navigation or object detection, is not explored.
(3)The paper introduces scene graphs as a key component of the Sg-CityU method to address the challenges of long-range connections and spatial inference in city scenes. However, the analysis of the impact of scene graphs on the model's performance is limited. A more detailed ablation study or visualization of the scene graph's role in the reasoning process could help readers better understand the effectiveness of this approach and its potential limitations.

**Suitability:**

3

---

### Official Review · Reviewer_vVH7 · 2024-05-24

**Rating:** 4
**Confidence:** 3

**Summary:**

In this paper, the authors point out the issue of lacking spatial semantic information and city-level interaction information in 3D MQA tasks. To solve these questions, the authors devote themselves to city-level sense understanding. Based on City-3DQA, a large-scale dataset considering scene semantic information, the authors propose Sg-CityU to generate high-quality city-related answers with spatial relationship information. They also introduce a new benchmark to evaluate the City-3DQA task. Experiments show the effectiveness of Sg-CityU in robustness and generalization.

**Strengths:**

1. The paper is well-written and easy to understand. The authors focus on a novel setting of the 3D-MQA task, which is practical in real applications. The paper is well-motivated and solves some crucial problems in the city-level QA task and the proposed task setting is worth doing future research on.
2. The paper proposes a new dataset containing city-level point clouds, QA, and other important annotations, which will be useful for more future research.
3. The experimental results show the effectiveness of Sg-CityU.

**Limitations:**

1. The proposed framework of Sg-CityU lacks novelty. The components of it are just some simple network structures.
2. There should be more ablation studies about Sg-CityU. i.e. how about combining the scene graph with other models? And there should also be some experiments to show the effects of the MMFN module.
3. In Tab.3 and 4, top-10 results seem perfect and meaningless, for there is no differentiation in many settings. More detailed results(i.e. top3, top5) should be contained.

Questions:
1. According to the method pipeline introduced in the paper, should the dimension of the 2-dimensional variables $F_p$ and $F_{sg}$ be $N\times dim$?
2. Can you give more detailed information about the answer set $A$? i.e. is it a static set with some template answers or dynamic tensors? And how to generate the final answer through the answer set?
3. The limitation of the dataset should be discussed. It seems like the cities in City-3DQA are all located in China. Will this lead to some bias for cities in different countries or regions are really different.
4. I expect to see some error analyses of Sg-CityU in the future version for the top-1 accuracy is still not satisfying.

Minor:
1. typos: line 483 $n$->$N$

**Suitability:**

3

---

### Official Review · Reviewer_kmov · 2024-05-24

**Rating:** 6
**Confidence:** 3

**Summary:**

This paper focuses on developing a 3D question answering system tailored for city planning and construction. The system integrates 3D models with natural language processing to provide detailed and contextually relevant answers to complex urban planning questions.

**Strengths:**

1.Industry Relevance: 3D question answering systems have a strong demand in urban planning and construction, making it practical.
2.Technology Integration: The combination of 3D models and NLP is clever. Detailed Results: The results explore the impact of the system's performance on urban planning potential.

**Limitations:**

1.User Interaction: Few details regarding how users interact with the system and user interface information are provided.
2.Comparison Benchmark: This paper lacks a comparative analysis with other similar systems.

**Suitability:**

3

---

### Meta-Review · Area_Chair_wf9D · 2024-06-23

**Recommendation:** Accept (Poster)
**Confidence:** 5

**Metareview:**

A good paper with four positive reviews. The reviewers claim to be satisfied by the rebuttal. Overall there is an agreement about accepting the paper.